# Synthesis and Elimination Pathways of 1-Methanesulfonyl-1,2-dihydroquinoline Sulfonamides

**DOI:** 10.3390/molecules28073256

**Published:** 2023-04-05

**Authors:** Ebenezer Ametsetor, Kwabena Fobi, Richard A. Bunce

**Affiliations:** Department of Chemistry, Oklahoma State University, Stillwater, OK 74078-3071, USA; eametse@okstate.edu (E.A.); kfobi@okstate.edu (K.F.)

**Keywords:** Morita–Baylis–Hillman acetates, elimination, tertiary 1,2-dihydroquinoline sulfonamides, quinolines, 1,3-sulfonyl migration

## Abstract

A series of new Morita–Baylis–Hillman acetates were prepared and reacted with methanesulfonamide (K_2_CO_3_, DMF, 23 °C) to produce tertiary dihydroquinoline sulfonamides in high yields. Subsequent efforts to eliminate the methylsulfonyl group from these derivatives (K_2_CO_3_, DMF, 90 °C) as a route to quinolines were met with mixed results. Although dihydroquinoline sulfonamides prepared from ethyl acrylate and acrylonitrile generally underwent elimination to give excellent yields of quinolines, those generated from 3-buten-2-one failed to undergo elimination and instead decomposed. The failure of these ketone substrates to aromatize presumably derives from the enolizable methyl ketone at C-3. Finally, the attempted aromatization of the acrylate-derived 6,7-difluoro-1,2-dihydroquinoline sulfonamide demonstrated that other interesting processes could occur in preference to the desired elimination.

## 1. Introduction

We recently reported the synthesis of quinolines, naphthyridines and their dihydro derivatives from Morita–Baylis–Hillman (MBH) acetates [1,2]. In the current work, we hoped to develop another approach to these structures involving the reaction of sulfonamides with these same activated substrates. There is ample precedent showing that sulfonamides can be used in aza–Michael reactions to prepare simple adducts as well as a variety of heterocycles [3,4,5,6,7,8,9,10]. Additionally, we showed in the past that aromatic sulfonamide anions readily undergo S_N_Ar additions to activated aromatic halides [11]. Our plan sought to utilize an aza–Michael-initiated S_N_2’-S_N_Ar sequence between methanesulfonamide (pKa 17.5 in DMSO [12]) or *p*-toluenesulfonamide (pKa ca. 16.1 in DMSO [13]) and an MBH acetate to generate a tertiary dihydroquinoline sulfonamide. Subsequent elimination of the sulfonyl group would then yield substituted quinoline products. For the purpose of atom economy [14,15,16], the elimination of a methylsulfonyl (mesyl) group was deemed to be preferable for this reaction.

The original project aimed to add methanesulfonamide to MBH acetates **1** derived from benzaldehydes activated by electron withdrawing groups (EWG) toward S_N_Ar cyclization via Michael-initiated S_N_2’ intermediate **A** and Meisenheimer complex **B** to give tertiary cyclic sulfonamides **2**. In the second step, the plan was to eliminate the mesyl group to generate 3,6-disubstituted quinoline derivatives (Figure 1). Such eliminations were first observed by Kim and co-workers [17] from 1-*p*-tosyl-1,2-dihydroquinolines derived from *p*-tosylamido MBH adducts. During our study, however, it was found that this process only proceeded using dihydroquinolines incorporating C-3 esters and nitriles but failed when a C-3 acetyl group was present.

A previous study by other researchers described a similar ring-forming process involving a S_N_2’-copper catalyzed coupling process using *p*-toluene- and *p*-nitrophenylsul-fonamides [18] with MBH acetates activated by an acrylate ester (Figure 2). This procedure did not require S_N_Ar-activating substituents on the aromatic ring of the MBH acetate, but instead, it coupled the initially added sulfonamide nitrogen to a halogenated carbon at the ortho position. This method yielded a diverse selection of substituted dihydroquinolines that were eliminated (4 equiv. DBU, CH_3_CN, 80 °C) to give ethyl 3-quinolinecarboxylates. Though this sequence shows a broader scope than the current work, reactions involving metal-promoted couplings can often be difficult to purify.

The formation of sulfonamides as intermediates in the current reaction is also of interest, as this family of compounds constitutes an important class of drug compounds with wide-ranging activities. Several experimental drug compounds are pictured in Figure 1. The tetrahydroquinoline sulfonamide **7** was found to impede the development and progression of several cancers by inhibiting the antagonist activity of myeloid cell leukemia-1 toward pro-apoptotic Bcl-2 proteins [19]. Phenylpyrrolidine sulfonamide **8** in conjunction with Doxorubicin showed antiproliferative activity against breast cancer cells both in vitro and in xenograft models [20]. Indole sulfonamide **9** inhibited the carbonic anhydrase IX isoform, possessed anti-proliferitive potential, and significantly increased apoptosis in breast cancer cells [21]. Coumarin-proline sulfonamide **10** proved to be active against breast cancer cells, and its inhibition of dipeptidal peptidase IV suggested that compound hybrids incorporating this functional triad might have potential in the treatment of type II diabetes [22]. Heteroaryl sulfonamide **11** exhibited neuroprotective effects via the inhibition of human carbonic anhydrase I and II, which may yield a new strategy for the treatment of Alzheimer’s disease [23]. Finally, pyrrolidinediol **12** proved effective in reversing pulmonary edema and increasing arterial oxygenation, lowering the chances of death in congestive heart failure patients through antagonism of the transient receptor potential vallinoid-4 (TRPV4) cation channel [24]. Although most sulfonamides are prepared via reaction of a sulfonyl chloride with a primary or secondary amine or alkylation of a pre-existing sulfonamide [25], the scheme outlined in this report represents a unique variation on double alkylation of a primary sulfonamide.

## 2. Results and Discussion

The MBH acetates were prepared as previously described [1,2]. The reaction of 2-fluoro-C5-activated benzaldehydes with ethyl acrylate, acrylonitrile, or 3-buten-2-one in the presence of DABCO in dry CH_3_CN gave the MBH alcohols, which were not isolated. Subsequent treatment of the adducts with acetic anhydride in the presence of catalytic trimethylsilyl trifluoromethanesulfonate [26] gave acetates **13**–**27** for the current study (Table 1). These substrates were spectroscopically pure and used directly without further purification.

Our results are summarized in Table 2. For most substrates, the initial addition of methanesulfonamide (and in one case, of *p*-toluenesulfonamide) with ester and ketone substrates (1:1:2 methanesulfonamide: MBH acetate: K_2_CO_3_, DMF, 23 °C, 2 h) occurred to produce 1-methanesulfonyl-1,2-dihydroquinolines in high yields. Additionally, the *p*-toluenesulfonyl derivative **28**-**Ts** was prepared in similar yields to demonstrate that aromatic sulfonamides could also be used in the ring closure and elimination protocols. Acrylonitrile substrates **35**–**37**, however, failed to undergo this reaction, presumably due to the diminished activation of the S_N_Ar acceptor ring by CN, CF_3_ and Cl relative to NO_2_. Attempts to isolate the intermediate Michael adducts were thwarted by the presence of multiple products, and prolonged heating led to eventual decomposition.

Our subsequent attempts to perform elimination reactions to produce quinoline derivatives met with varied results (Table 3). With one exception, eliminations from acrylate-derived substrates were successful in producing the targeted quinolines in high yields (1 equiv. K_2_CO_3_, DMF, 90 °C, 1–2 h). Additionally, the nitro-substituted acrylonitrile adduct gave excellent conversion to the heteroaromatic product, whereas other substrates incorporating methyl ketones decomposed under these conditions. We note that attempts to perform the elimination with DBU as reported earlier [18] also failed for these substrates. Apparently, this aromatization becomes problematic when enolizable protons reside on the C-3 substituent of the sulfonyl-bearing ring. Eliminations to form naphthyridines were also unsuccessful. Thus, the limitations of the reaction appear to be reasonably well-defined.

To further explore functionalization of these structures, we wished to evaluate 6,7-difluorodihydroquinoline **32** in a possible extended S_N_Ar reaction [27] through the fused-ring system. Although the carboxylic ester is a weak activator for S_N_Ar reactions, our initial expectation was that the C-7 fluorine might be displaced to give the phenethylamino residue at this position. A similar substitution is a key step in the synthesis of fluoroquinolone antibiotics such as ciprofloxacin [28]. When the 6,7-difluoro substrate was reacted with phenethylamine, however, an unexpected product was observed in 95% yield. ^1^H NMR indicated that phenethylamine had been incorporated into the product. Additionally, the nitrogen-containing ring had been aromatized, and the methyl singlet of the mesyl in the starting material, though shifted from δ 2.71 in **32** to δ 2.90, was still present. Finally, the lack of ^19^F-coupled signals in the ^13^C NMR confirmed that the product contained no fluorine. An X-ray structure was obtained to unequivocally reveal the positions of the various groups. Interestingly, the structure proved to be the 1,3-sulfonyl migrated amine **49** (Figure 2), the C-6 fluorine having been replaced by a phenethylamino group and the C-7 fluorine replaced by a mesyl group. Reference to the literature indicated that 1,3-sulfonyl migrations are common but usually catalyzed by transition metals or Lewis acids [29]. Only one example was found where sulfonyl migration occurred under conditions similar to those reported here, and this involved a base-catalyzed aza–Claisen rearrangement of an *N*-arylsulfonyl propargyl vinylamine [30]. In the current work, the rearrangement was performed using 1:1:1 phenethylamine: **32**: K_2_CO_3_ in DMF at 90 °C; no transition metals or Lewis acids were present.

Two mechanisms were considered to rationalize the observed transformation. Mechanism 1 required S_N_Ar addition–elimination through two rings, whereas mechanism 2 avoided this extended S_N_Ar process. In mechanism 1 (Figure 3), phenethylamine would attack the C-6 position with the facile elimination of the mesyl group to give **C**. This process is critical to the current transformation, as the *N*-benzyl analogue of **32** was inert to these same conditions. Deprotonation by base would subsequently aromatize this intermediate to give **D**. At this point, the mesyl anion could attack at C-7 in a S_N_Ar-type addition to give **E**. Final elimination of the C-7 fluoride would then be expected to yield the observed product, **49**. The weakness of this mechanism is that the ethoxycarbonyl group does not provide sufficient activation for the extended S_N_Ar reaction. Additionally, the phenethylamino group at C-6 would be expected to deactivate the ring toward this addition process.

Mechanism 2 (Figure 4) begins with the formation of intermediate **C** as described previously. In the subsequent step, however, the sulfonyl anion would add to C-7 to give the imine stabilized anion **F**, which could eliminate the C-7 fluoride to give intermediate **G**. Finally, the base-promoted rearomatization of the system would afford product **49**. Although both mechanisms traverse several high energy intermediates, we tend to favor this pathway, as it appears to have better electronic features than mechanism 1. At this point, however, we cannot definitively state which mechanism is operating.

## 3. Methods and Materials

### 3.1. General Methods

All reactions were performed under dry N_2_ in oven-dried glassware. All reagents and solvents were used as received. All wash solutions in work-up procedures were aqueous. Reactions were monitored with thin layer chromatography on Analtech No 21521 silica gel GF plates (Newark, DE, USA). Preparative separations were performed either by using flash chromatography on silica gel (Davisil^®^, grade 62, 60–200 mesh) containing 0.5% of UV-05 UV-active phosphor (both from Sorbent Technologies, Norcross, GA, USA) slurry packed into quartz columns or by using preparative thin layer chromatography (PTLC) on Analtech No 02,015 silica gel GF plates (Newark, DE, USA). Band elution for all chromatographic separations was monitored using a handheld UV lamp (Fisher Scientific, Pittsburgh, PA, USA). Melting points were obtained using a MEL-TEMP apparatus (Cambridge, MA, USA) and are uncorrected. FT-IR spectra were run as thin films on NaCl disks using a Nicolet iS50 spectrophotometer (Madison, WI, USA). ^1^H- and ^13^C-NMR spectra were measured using a Bruker Avance 400 system (Billerica, MA, USA) at 400 MHz and 101 MHz, respectively, using the indicated solvents containing 0.05% (CH_3_)_4_Si as the internal standard; coupling constants (*J*) are given in Hz. Low-resolution mass spectra were obtained using a Hewlett-Packard Model 1800A GCD GC-MS system (Palo Alto, CA, USA). Elemental analyses (±0.4%) were determined for all new compounds by Atlantic Microlabs (Norcross, GA, USA).

### 3.2. General Procedure for the Preparation of MBH Acetates

To a stirred solution of DABCO (1 equiv.) in CH_3_CN at room temperature, ethyl acrylate, acrylonitrile or 3-buten-2-one (1 equiv.) was added. The aldehyde was then added, and stirring continued until TLC analysis (10% EtOAc/hexane) indicated that reaction was complete (12–48 h). The crude reaction mixture was poured into water (30 mL) and extracted with EtOAc (3 × 25 mL). The combined organic layers were washed with 1 M HCl (2 × 25 mL), saturated NaHCO_3_ (30 mL) and saturated NaCl (30 mL) and then dried (Na_2_SO_4_). Concentration under vacuum gave the MBH alcohol, which was used without further purification.

The general procedure of Procopiou and co-workers was used [26]. A solution of the MBH alcohol (1 mmol) in CH_2_Cl_2_ (2 mL) was treated with acetic anhydride (1.5 mmol) at 0 °C, followed by a solution of TMSOTf (1.0 M in CH_2_Cl_2_, 20 µL). The reaction mixture was stirred for 1 h, after which TLC (10% EtOAc/hexane) indicated the reaction was complete. A stoichiometric quantity of methanol (142 µL) was added to quench the excess acetic anhydride, and the crude reaction mixture was washed with saturated NaHCO_3_ (30 mL). The two phases were separated, and the aqueous layer was further extracted with CH_2_Cl_2_ (2 × 25 mL). The combined organic extracts were washed with saturated NaHCO_3_ (30 mL), water (30 mL) and saturated NaCl (30 mL) and then dried (Na_2_SO_4_). The solvent was removed under vacuum to afford the spectroscopically pure MBH acetate, which was used directly for the preparation of 1-methylsulfonyl-1,2-dihydroquinolines. (±)-Ethyl 2-(acetoxy(2-fluoro-5-nitrophenyl)methyl)acrylate (**13**), (±)-ethyl 2-(acetoxy(5-cyano-2-fluorophenyl)methyl)acrylate (**14**), (±)-ethyl 2-(acetoxy(2-fluoropyridin-3-yl)methyl)acrylate (**18**), (±)-2-cyano-1-(2-fluoro-5-nitrophenyl)allyl acetate (**19**) and (±)-2-cyano-1-(5-cyano-2-fluorophenyl)allyl acetate (**20**) were prepared by using this procedure and were characterized previously [2].

#### 3.2.1. (±)-Ethyl 2-(acetoxy(2-fluoro-5-(trifluoromethyl)phenyl)methyl)acrylate (**15**)

Yield: 2.10 g (92%) as a yellow oil; IR: 1756, 1724, 1620 cm^−1^; ^1^H NMR (400 MHz, CDCl_3_): δ 7.64 (dd, *J* = 6.5, 2.5 Hz, 1H), 7.59 (m, 1H), 7.20 (t, *J* = 9.0 Hz, 1H), 6.96 (s, 1H), 6.52 (s, 1H), 5.91 (s, 1H), 4.17 (q, *J* = 7.1 Hz, 2H), 2.14 (s, 3H), 1.24 (t, *J* = 7.1 Hz, 3H); ^13^C NMR (101 MHz, CDCl_3_): δ 169.1, 146.4, 162.1 (d, *J* = 256.0 Hz), 137.9, 127.7–127.4 (complex), 127.0, 126.6, 126.6–126.5 (complex), 123.6 (q, *J* = 272.1 Hz), 119.5, 116.6, 116.3, 66.9, 20.7, 13.8; MS (*m/z*): 334 (M^+·^); Anal. Calcd for C_15_H_14_F_4_O_4_: C, 53.90; H, 4.22. Found: C, 53.81; H, 4.18.

#### 3.2.2. (±)-Ethyl 2-(acetoxy(5-chloro-2-fluorophenyl)methyl)acrylate (**16**)

Yield: 2.16 g (93%) as a yellow oil; IR: 1753, 1725, 1637 cm^−1^; ^1^H NMR (400 MHz, CDCl_3_): δ 7.30 (m, 1H), 7.25 (m, 1H), 7.00 (t, *J* = 8.9 Hz, 1H), 6.87 (s, 1H), 6.48 (s, 1H), 5.85 (s, 1H), 4.17 (q, *J* = 7.1 Hz, 2H), 2.13 (s, 3H), 1.24 (t, *J* = 7.1 Hz, 3H); ^13^C NMR (101 MHz, CDCl_3_): δ 169.0, 164.4, 158.8 (d, *J* = 250.5 Hz), 138.1, 130.0 (d, *J* = 8.5 Hz), 129.2 (d, *J* = 3.4 Hz), 128.8 (d, *J* = 3.6 Hz), 127.1 (d, *J* = 15.1 Hz), 126.9, 117.1 (d, *J* = 23.6 Hz), 66.9 (d, *J* = 2.9 Hz), 61.2, 20.8, 13.9; MS (*m/z*): 300 (M^+^); Anal. Calcd for C_14_H_14_ClFO_4_: C, 55.92; H, 4.69. Found: C, 55.84; H, 4.66.

#### 3.2.3. (±)-Ethyl 2-(acetoxy(2,4,5-trifluorophenyl)methyl)acrylate (**17**)

Yield: 2.13 g (92%) as a yellow oil; IR: 1757, 1728, 1634 cm^−1^; ^1^H NMR (400 MHz, CDCl_3_): δ 7.17 (ddd, *J* = 15.1, 8.8, 6.5 Hz, 1H), 6.94 (td, *J* = 10.3, 6.5 Hz, 1H), 6.84 (s, 1H), 6.48 (s, 1H), 5.88 (s, 1H), 4.18 (q, *J* = 7.1 Hz, 2H), 2.13 (d, *J* = 1.4 Hz, 3H), 1.25 (td, *J* = 7.1, 1.4 Hz, 3H); ^13^C NMR (101 MHz, CDCl_3_): δ 169.0, 164.3, 155.4 (dd, *J* = 249.4, 2.7 Hz), 150.0 (ddd, *J* = 252.5, 14.4, 12.2 Hz), 147.2 (dd, *J* = 245.4, 3.7 Hz), 138.0, 126.6, 121.9 (dt, *J* = 15.8, 4.7 Hz), 116.7 (ddd, *J* = 20.2, 4.7, 1.6 Hz), 105.8 (dd, *J* = 27.9, 20.9 Hz), 66.6 (d, *J* = 2.0 Hz), 61.2, 20.7, 13.9: MS (*m/z*): 302 (M^+^); Anal. Calcd for C_14_H_13_F_3_O_4_: C, 55.63; H, 4.34. Found: C, 55.68; H, 4.39.

#### 3.2.4. (±)-2-Cyano-1-(2-fluoro-5-(trifluoromethyl)phenyl)allyl Acetate (**21**)

Yield: 1.09 g (93%) as a white solid, m.p. 35–36 °C; IR: 2229, 1761, 1626 cm^−1^; ^1^H NMR (400 MHz, CDCl_3_): δ 7.79 (dd, *J* = 6.5, 2.3 Hz, 1H), 7.67 (m, 1H), 7.24 (t, *J* = 9.0 Hz, 1H), 6.64 (s, 1H), 6.15 (d, *J* = 1.0 Hz, 1H), 6.12 (t, *J* = 1.0 Hz, 1H), 2.23 (s, 3H); ^13^C NMR (101 MHz, CDCl_3_): δ 168.9, 161.5 (d, *J* = 255.5 Hz), 133.3, 128.4 (dq, *J* = 9.6, 3.7 Hz), 127.6 (qd, *J* = 33.4, 3.7 Hz), 125.5 (apparent pentet, *J* = 3.9 Hz), 124.5 (d, *J* = 14.0 Hz), 123.4 (q, *J* = 272.2 Hz), 121.3, 116.7 (d, *J* = 22.5 Hz), 115.4, 68.0 (d, *J* = 3.3 Hz), 20.8; MS (*m/z*): 287 (M^+^); Anal. Calcd for C_13_H_9_F_4_NO_2_: C, 54.36; H, 3.16; N, 4.88. Found: C, 54.41; H, 3.11; N, 4.83.

#### 3.2.5. (±)-1-(5-Chloro-2-fluorophenyl)-2-cyanoallyl Acetate (**22**)

Yield: 1.14 g (95%) as a white solid, m.p. 45–46 °C; IR: 2231, 1756, 1628 cm^−1^; ^1^H NMR (400 MHz, CDCl_3_): δ 7.48 (dd, *J* = 6.2, 2.6 Hz, 1H), 7.34 (ddd, *J* = 8.8, 4.5, 2.6 Hz, 1H), 7.05 (t, *J* = 9.1 Hz, 1H), 6.57 (s, 1H), 6.13 (d, *J* = 1.0 Hz, 1H), 6.09 (t, *J* = 1.0 Hz, 1H), 2.22 (s, 3H); ^13^C NMR (101 MHz, CDCl_3_): δ 168.9, 158.2 (d, *J* = 249.1 Hz), 133.1, 130.9 (d, *J* = 8.5 Hz), 130.1 (d, *J* = 3.5 Hz), 127.8 (d, *J* = 3.1 Hz), 125.0 (d, *J* = 14.6 Hz), 121.4, 117.3 (d, *J* = 23.0 Hz), 115.6 68.0 (d, *J* = 3.3 Hz), 20.8; MS (*m/z*): 253 (M^+^); Anal. Calcd for C_12_H_9_ClFNO_2_: C, 56.82; H, 3.58; N, 5.52. Found: C, 56.70; H, 3.53; N, 5.42.

#### 3.2.6. (±)-1-(2-Fluoro-5-nitrophenyl)-2-methylene-3-oxobutyl Acetate (**23**)

Yield: 2.68 g (96%) as a light yellow solid, m.p. 72–73 °C; IR: 1752, 1679, 1636 cm^−1^; ^1^H NMR (400 MHz, CDCl_3_): δ 8.22–8.18 (complex, 2H), 7.21 (t, *J* = 9.5 Hz, 1H), 6.93 (s, 1H), 6.32 (d, *J* = 0.8 Hz, 1H), 6.02 (d, *J* = 1.4 Hz, 1H), 2.37 (s, 3H), 2.14 (s, 3H); ^13^C NMR (101 MHz, CDCl_3_): δ 196.9, 169.0, 163.8 (d, *J* = 261.8 Hz), 145.3, 144.2 (d, *J* = 3.0 Hz), 127.7 (d, *J* = 15.5 Hz), 126.3, 125.9 (d, *J* = 10.4 Hz), 125.0 (d, *J* = 5.5 Hz), 116.9 (d, *J* = 24.5 Hz), 66.3 (d, *J* = 2.9 Hz), 26.0, 20.8; MS (*m/z*): 281 (M^+^); Anal. Calcd for C_13_H_12_FNO_5_: C, 55.52; H, 4.30; N, 4.98. Found: C, 55.46; H, 4.22; N, 4.91.

#### 3.2.7. (±)-1-(5-Cyano-2-fluorophenyl)-2-methylene-3-oxobutyl Acetate (**24**)

Yield: 1.17 g (98%) as a white solid, m.p. 104–105 °C; IR: 2230, 1754, 1681, 1627 cm^−1^; ^1^H NMR (400 MHz, CDCl_3_): δ 7.64 (dd, *J* = 7.0, 2.1 Hz, 1H), 7.61 (m, 1H), 7.16 (t, *J* = 9.0 Hz, 1H), 6.89 (s, 1H), 6.35 (s, 1H), 6.15 (d, *J* = 1.4 Hz, 1H), 2.36 (s, 3H), 2.13 (s, 3H); ^13^C NMR (101 MHz, CDCl_3_): δ 196.8, 169.0, 162.7 (d, *J* = 260.4 Hz), 145.4, 134.3 (d, *J* = 9.7 Hz), 133.6 (d, *J* = 4.9 Hz), 127.8 (d, *J* = 14.7 Hz), 127.1, 117.9, 117.3 (d, *J* = 23.4 Hz), 108.6 (d, *J* = 4.0 Hz), 66.4 (d, *J* = 3.0 Hz), 26.0, 10.9; HRMS (*m/z*): 261 (M^+^); Anal. Calcd for C_14_H_12_FNO_3_: C, 64.36; H, 4.63; N, 5.36. Found: C, 64.29; H, 4.58; N, 5.27.

#### 3.2.8. (±)-1-(2-Fluoro-5-(trifluoromethyl)phenyl)-2-methylene-3-oxobutyl Acetate (**25**)

Yield: 3.19 g (95%) as a yellow oil; IR: 1749, 1684, 1631 cm^−1^; ^1^H NMR (400 MHz, CDCl_3_): δ 7.57 (overlapping d, *J* = 5.3 Hz, 1H and m, 1H), 7.18 (t, *J* = 9.3 Hz, 1H), 6.96 (s, 1H), 6.35 (s, 1H), 6.12 (s, 1H), 2.35 (s, 3H), 2.12 (s, 3H); ^13^C NMR (101 MHz, CDCl_3_): δ 196.9, 169.1, 162.1 (dd, *J* = 255.7, 1.6 Hz), 145.7, 127.5 (m, 2C), 126.9 (d, *J* = 2.0 Hz), 126.8–126.4 (complex, 2C), 123.6 (q, *J* = 271.9 Hz), 116.5 (d, *J* = 23.2 Hz), 66.7 (d, *J* = 2.9 Hz), 26.0, 20.8; MS (*m/z*): 304 (M^+·^); Anal. Calcd for C_14_H_12_F_4_O_3_: C, 55.27; H, 3.98. Found: C, 55.21; H, 3.96.

#### 3.2.9. (±)-2-Methylene-3-oxo-1-(2,4,5-trifluorophenyl)butyl Acetate (**26**)

Yield: 2.98 g (97%) as a white solid, m.p. 82–84 °C; IR: 1754, 1680, 1631 cm^−1^; ^1^H NMR (400 MHz, CDCl_3_): δ 7.14 (ddd, *J* = 15.2, 8.7, 6.5 Hz, 1H), 6.94 (td, *J* = 9.7, 6.5 Hz, 1H), 6.84 (s, 1H), 6.33 (s, 1H), 6.12 (s, 1H), 2.35 (s, 3H), 2.11 (s, 3H); ^13^C NMR (101 MHz, CDCl_3_): δ 196.9, 169.1, 155.5 (ddd, *J* = 249.4, 9.5, 2.7 Hz), 150.0 (ddd, *J* = 252.3, 14.3, 12.2 Hz), 146.7 (ddd, *J* = 245.1, 12.7, 3.7 Hz), 146.5, 126.7 (d, *J* = 1.2 Hz), 122.1 (dt, *J* = 15.6, 4.7 Hz), 117.0 (ddd, *J* = 20.2, 5.1, 1.5 Hz), 105.9 (dd, *J* = 27.7, 20.9 Hz), 66.5 (d, *J* = 2.4 Hz), 26.0, 20.8; MS (*m/z*): 272 (M^+^); Anal. Calcd for C_13_H_11_F_3_O_3_: C, 57.36; H, 4.07. Found: C, 57.31; H, 3.98.

#### 3.2.10. (±)-1-(2-Fluoropyridin-3-yl)-2-methylene-3-oxobutyl Acetate (**27**)

Yield: 2.16 g (89%) as a yellow oil; IR: 1749, 1679, 1638 cm^−1^; ^1^H NMR (400 MHz, CDCl_3_): δ 8.15 (dm, *J* = 4.9 Hz, 1H), 7.82 (ddd, *J* = 9.5, 7.4, 2.0 Hz, 1H), 7.19 (ddd, *J* = 6.6, 4.9, 1.6 Hz, 1H), 6.81 (s, 1H), 6.37 (s, 1H), 6.19 (s, 1H), 2.35 (s, 3H), 2.12 (s, 3H); ^13^C NMR (101 MHz, CDCl_3_): δ 197.0, 169.2, 160.9 (d, *J* = 241.9 Hz), 147.4 (d, *J* = 14.8 Hz), 145.1 140.3 (*J* = 4.4 Hz), 127.3 (d, *J* = 1.7 Hz), 121.4 (d, *J* = 4.5 Hz), 120.6 (d, *J* = 27.5 Hz), 67.7 (d, *J* = 2.8 Hz), 26.0, 20.8; MS (*m/z*): 237 (M^+^); Anal. Calcd for C_12_H_12_FNO_3_: C, 60.76; H, 5.10; N, 5.90. Found: C, 60.69; H, 5.07; N, 5.81.

### 3.3. General Procedure for the Preparation of 1-Methanesulfonyl-1,2-Dihydroquinolines from MBH Acetates

To a mixture of methane- or *p*-toluenesulfonamide (1 mmol) and K_2_CO_3_ (2 mmol) in anhydrous DMF (1 mL), the MBH acetate (1 mmol) was added in DMF (1 mL). The mixture was stirred at room temperature for 2 h, at which time TLC (20% EtOAc/hexane) indicated the complete consumption of the starting material. The crude reaction mixture was poured into water (15 mL) and extracted with EtOAc (3 × 25 mL). The combined organic layers were washed with 1 M HCl (2 × 25 mL), saturated NaHCO_3_ (2 × 25 mL) and saturated NaCl (1 × 30 mL) and then dried (Na_2_SO_4_). Removal of the solvent under vacuum gave a crude product that was further purified via column chromatography to afford the following products.

#### 3.3.1. Ethyl 1-(Methylsulfonyl)-6-nitro-1,2-dihydroquinoline-3-carboxylate (**28-Ms**)

Yield: 2.01 g (96%) as a white solid, m.p. 163–164 °C; IR: 1704, 1642, 1529, 1352, 1160 cm^−1^; ^1^H NMR (400 MHz, CDCl_3_): δ 8.24–8.17 (complex, 2H), 7.84 (d, *J* = 8.8 Hz, 1H), 7.60 (s, 1H), 4.75 (d, *J* = 1.5 Hz, 2H), 4.35 (q, *J* = 7.1 Hz, 2H), 2.85 (s, 3H), 1.39 (t, *J* = 7.1 Hz, 3H); ^13^C NMR (101 MHz, CDCl_3_): δ 163.7, 145.4, 141.6, 132.4, 128.6, 127.6, 125.5, 125.4, 124.0, 61.8, 44.5, 39.2, 14.2; MS (*m/z*): 326 (M^+·^); Anal. Calcd for C_13_H_14_N_2_O_6_S: C, 47.85; H, 4.32; N, 8.58. Found: C, 47.92; H, 4.34; N, 8.51.

#### 3.3.2. Ethyl 6-Nitro-1-tosyl-1,2-dihydroquinoline-3-carboxylate (**28**-**Ts**)

Yield: 2.35 g (91%) as a white solid, m.p. 142–143 °C (lit. [18] m.p. 140–142 °C); IR 1708, 1524, 1349, 1167 cm^−1^; ^1^H NMR (400 MHz, CDCl_3_): δ 8.23 (dd, *J* = 9.0, 2.6 Hz, 1H), 8.02 (d, *J* = 2.6 Hz, 1H), 7.93 (d, *J* = 9.0 Hz, 1H), 7.36 (d, *J* = 8.3 Hz, 2H), 7.13 (d, *J* = 8.3 Hz, 2H), 7.07 (s, 1H), 4.75 (s, 2H), 4.26 (q, *J* = 7.1 Hz, 2H), 2.36 (s, 3H), 1.34 (t, *J* = 7.1 Hz, 3H); ^13^C NMR (101 MHz, CDCl_3_): δ 163.5, 145.6, 144.7, 141.7, 135.6, 131.5, 129.7, 128.3, 127.8, 127.0, 126.9, 125.0, 123.5, 61.4, 44.3, 21.6, 14.3; MS (*m/z*): 402 (M^+^).

#### 3.3.3. Ethyl 6-Cyano-1-(methylsulfonyl)-1,2-dihydroquinoline-3-carbonitrile (**29**)

Yield: 0.49 g (93%) as a white solid, m.p. 103–104 °C; IR: 2226, 1707, 1645, 1348, 1151 cm^−1^; ^1^H NMR (400 MHz, CDCl_3_): δ 7.79 (d, *J* = 8.5 Hz, 1H), 7.65 (dd, *J* = 8.4, 2.0 Hz, 1H), 7.60 (d, *J* = 2.0 Hz, 1H), 7.51 (s, 1H), 4.72 (d, *J* = 1.4 Hz, 2H), 4.34 (q, *J* = 7.1 Hz, 2H), 2.80 (s, 3H), 1.38 (t, *J* = 7.1 Hz, 3H); ^13^C NMR (101 MHz, CDCl_3_): δ 163.8, 140.0, 133.9, 132.5, 132.2, 128.5, 127.9, 125.8, 117.7, 110.2, 61.8, 44.4, 39.0, 14.2; MS (*m/z*): 306 (M^+^); Anal. Calcd for C_14_H_14_N_2_O_4_S: C, 54.89; H, 4.61; N, 9.14. Found: C, 54.84; H, 4.57; N, 9.03.

#### 3.3.4. Ethyl 1-(Methylsulfonyl)-6-(trifluoromethyl)-1,2-dihydroquinoline-3-carboxylate (**30**)

Yield: 0.40 g (96%) as a white solid, m.p. 99–101 °C; IR: 1714, 1648, 1339, 1130 cm^−1^; ^1^H NMR (400 MHz, CDCl_3_): δ 7.80 (d, *J* = 8.5 Hz, 1H), 7.63 (dd, *J* = 8.5, 2.2 Hz, 1H), 7.58 (two coincident signals, apparent s, 2H), 4.72 (d, *J* = 1.4 Hz, 2H), 4.33 (q, *J* = 7.1 Hz, 2H), 2.76 (s, 3H), 1.37 (t, *J* = 7.1 Hz, 3H); ^13^C NMR (101 MHz, CDCl_3_): δ 164.0, 139.1, 133.0, 128.8 (q, *J* = 33.1 Hz), 128.0, 127.6, 127.5 (q, *J* = 3.6 Hz), 126.0, 125.9 (q, *J* = 3.8 Hz), 123.5 (q, *J* = 272.2 Hz), 61.6, 44.4, 38.7, 14.2; MS (*m/z*): 349 (M^+^); Anal. Calcd for C_14_H_14_F_3_NO_4_S: C, 48.14; H, 4.04; N, 4.01. Found: C, 48.09; H, 3.99; N, 3.92.

#### 3.3.5. Ethyl 6-Chloro-1-(methylsulfonyl)-1,2-dihydroquinoline-3-carboxylate (**31**)

Yield: 0.38 g (91%) as a white solid, m.p. 109–111 °C; IR 1711, 1359, 1156 cm^−1^; ^1^H NMR (400 MHz, CDCl_3_): δ 7.61 (d, *J* = 8.6 Hz, 1H), 7.50 (s, 1H), 7.35 (dd, *J* = 8.6, 2.4 Hz, 1H), 7.32 (d, *J* = 2.4 Hz, 1H), 4.67 (d, *J* = 1.1 Hz, 2H), 4.32 (q, *J* = 7.1 Hz, 2H), 2.69 (s, 3H), 1.37 (t, *J* = 7.1 Hz, 3H); ^13^C NMR (101 MHz, CDCl_3_): δ 164.2, 134.5, 133.1, 132.5, 130.6, 128.8, 128.5, 127.9, 127.6, 61.6, 44.5, 38.1, 14.2; MS (*m/z*): 315. (M^+^); Anal. Calcd for C_13_H_14_ClNO_4_S: C, 49.45; H, 4.47; N, 4.44. Found: C, 49.41; H, 4.46; N, 4.35.

#### 3.3.6. Ethyl 6,7-Difluoro-1-(methylsulfonyl)-1,2-dihydroquinoline-3-carboxylate (**32**)

Yield: 0.38 g (93%) as a white solid, m.p. 145–147 °C; IR: 1718, 1659, 1354, 1155 cm^−1^; ^1^H NMR (400 MHz, CDCl_3_): δ 7.55 (dd, *J* = 11.2, 7.3 Hz, 1H), 7.47 (s, 1H), 7.15 (dd, *J* = 9.8, 8.2 Hz, 1H), 4.66 (d, *J* = 1.4 Hz, 2H), 4.35 (q, *J* = 7.1 Hz, 2H), 2.71 (s, 3H), 1.37 (t, *J* = 7.1 Hz, 3H); ^13^C NMR (101 MHz, CDCl_3_): δ 164.1, 150.8 (dd, *J* = 254.3, 13.1 Hz), 149.0 (dd, *J* = 250.0, 13.3 Hz), 132.6 (dd, *J* = 9.1, 3.3 Hz), 132.5 (t, *J* = 2.3 Hz), 127.0 (d, *J* = 2.6 Hz), 124.2 (dd, *J* = 6.6, 3.8 Hz), 116.7 (dd, *J* = 18.5, 1.5 Hz), 116.4 (d, *J* = 21.3 Hz), 61.6, 44.2, 38.2, 14.2; MS (*m/z*): 317 (M^+·^); Anal. Calcd for C_13_H_13_F_2_NO_4_S: C, 49.21; H, 4.13; N, 4.41. Found: C, 49.12; H, 4.08; N, 4.33.

#### 3.3.7. Ethyl 1-(Methylsulfonyl)-1,2-dihydro-1,8-naphthyridine-3-carboxylate (**33**)

Yield: 1.94 g (92%) as a white solid, m.p. 105–106 °C; IR: 1704, 1338, 1158 cm^−1^; ^1^H NMR (400 MHz, CDCl_3_): δ 8.28 (dd, *J* = 5.0, 1.8 Hz, 1H), 7.50 (dd, *J* = 7.5, 1.8 Hz, 1H), 7.39 (t, *J* = 1.6 Hz, 1H), 7.02 (dd, *J* = 7.5, 5.0 Hz, 1H), 4.84 (d, *J* = 1.6 Hz, 2H), 4.30 (q, *J* = 7.1 Hz, 2H), 3.51 (s, 3H), 1.35 (t, *J* = 7.1 Hz, 3H); ^13^C NMR (101 MHz, CDCl_3_): δ 164.0, 151.2, 148.7, 136.8, 132.5, 126.7, 119.6, 119.5, 61.4, 44.5, 42.3, 14.3; MS (*m/z*): 282 (M^+^); Anal. Calcd for C_12_H_14_N_2_O_4_S: C, 51.05; H, 5.00; N, 9.92. Found: C, 50.94; H, 4.96; N, 9.84.

#### 3.3.8. 1-(Methylsulfonyl)-6-nitro-1,2-dihydroquinoline-3-carbonitrile (**34**)

Yield: 0.79 g (96%) as a white solid, m.p. 126–127 °C; IR: 2231, 1627, 1529, 1355, 1329 cm^−1^; ^1^H NMR (400 MHz, CDCl_3_): δ 8.47 (d, *J* = 2.7 Hz, 1H), 8.41 (dd, *J* = 9.1, 2.7 Hz, 1H), 7.81 (s, 1H), 7.49 (d, *J* = 9.1 Hz, 1H), 5.11 (d, *J* = 1.4 Hz, 2H), 2.20 (s, 3H); ^13^C NMR (101 MHz, CDCl_3_): δ 170.2, 158.5, 156.8, 138.5, 126.5, 126.4, 123.8, 118.8, 117.9, 60.7, 20.8; MS (*m/z*): 279 (M^+^); Anal. Calcd for C_11_H_9_N_3_O_4_S: C, 47.31; H, 3.25; N, 15.05. Found: C, 47.25; H, 3.23; N, 14.96.

#### 3.3.9. 1-(1-(Methylsulfonyl)-6-nitro-1,2-dihydroquinolin-3-yl)ethan-1-one (**38**)

Yield: 3.26 g (93%) as a yellow solid, m.p. 195–196 °C; IR: 1677, 1524, 1344, 1169 cm^−1^; ^1^H NMR (400 MHz, CDCl_3_): δ 8.45 (d, *J* = 2.8 Hz, 1H), 8.25 (dd, *J* = 9.1, 2.8 Hz, 1H), 7.95 (s, 1H), 7.74 (d, *J* = 9.1 Hz, 1H), 4.58 (d, *J* = 1.3 Hz, 2H), 3.15 s, 3H), 2.44 (s, 3H); ^13^C NMR (101 MHz, CDCl_3_): δ 196.4, 144.5, 142.7, 135.3, 133.7, 128.0, 125.9, 125.1, 123.9, 43.5, 39.6, 25.7; MS (*m/z*): 296 (M^+^); Anal. Calcd for C_12_H_12_N_2_O_5_S: C, 48.64; H, 4.08; N, 9.45. Found: C, 48.59; H, 4.01; N, 9.37.

#### 3.3.10. 3-Acetyl-1-(methylsulfonyl)-1,2-dihydroquinoline-6-carbonitrile (**39**)

Yield: 3.26 g (88%) as a light yellow solid, mp 206–207 °C; IR: 2236, 1635, 1338, 1168 cm^−1^; ^1^H NMR (400 MHz, CDCl_3_): δ 7.79 (d, *J* = 8.4 Hz, 1H), 7.67 (dd, *J* = 8.4, 1.9 Hz, 1H), 7.64 (d, *J* = 1.9 Hz, 1H), 7.37 (s, 1H), 4.70 (d, *J* = 1.3 Hz, 2H), 2.82 (s, 3H), 2.48 (s, 3H); ^13^C NMR (101 MHz, CDCl_3_): δ 195.1, 140.5, 135.9, 134.3, 132.7, 132.1, 127.7, 125.4, 117.7, 109.9, 43.4, 39.1, 25.4; MS (*m/z*): 276 (M^+^); Anal. Calcd. for C_13_H_12_N_2_O_3_S: C, 56.51; H, 4.38; N, 10.14. Found: C, 56.48; H, 4.37; N, 10.05.

#### 3.3.11. 1-(1-(Methylsulfonyl)-6-(trifluoromethyl)-1,2-dihydroquinolin-3-yl)ethan-1-one (**40**)

Yield: 0.53 g (93%) as a white solid, m.p. 183–185 °C; IR: 1660, 1632, 1338, 1163 cm^−1^; ^1^H NMR (400 MHz, CDCl_3_): δ 7.80 (d, *J* = 8.4 Hz, 1H), 7.65 (dd, *J* = 10.6, 8.4 Hz, 1H), 7.62 (s, 1H), 7.45 (s, 1H), 4.70 (d, *J* = 1.3 Hz, 2H), 2.77 (s, 3H), 2.48 (s, 3H); ^13^C NMR (101 MHz, CDCl_3_): δ 195.3, 139.6, 135.6, 133.0, 128.6 (q, *J* = 33.4 Hz), 127.9 (q, *J* = 3.7 Hz), 127.4, 126.2 (q, *J* = 3.7 Hz), 125.6 123.5 (q, *J* = 272.2 Hz), 43.4, 38.8, 25.3; MS (*m/z*): 319 (M^+·^); Anal. Calcd for C_13_H_12_F_3_NO_3_S: C, 48.90; H, 3.79; N, 4.39. Found: C, 48.87; H, 3.76; N, 4.32.

#### 3.3.12. 1-(6,7-Difluoro-1-(methylsulfonyl)-1,2-dihydroquinolin-3-yl)ethan-1-one (**41**)

Yield: 0.35 g (84%) as a white solid, m.p. 173–174 °C; IR: 1670, 1646, 1345, 1163 cm^−1^; ^1^H NMR (400 MHz, CDCl_3_): δ 7.56 (dd, *J* = 11.3, 7.3 Hz, 1H), 7.33 (s, 1H), 7.19 (dd, *J* = 9.7, 8.2 Hz, 1H), 4.65 (s, 2H), 2.70 (s, 3H), 2.46 (s, 3H); ^13^C NMR (101 MHz, CDCl_3_): δ 195.2, 151.1 (dd, *J* = 255.0, 13.5 Hz), 148.8 (dd, *J* = 250.2, 13.4 Hz), 134.7 (d, *J* = 2.8 Hz), 133.2 (dd, *J* = 9.1, 3.3 Hz), 132.4 (t, *J* = 2.2 Hz), 124.0 (dd, *J* = 10.0, 3.6 Hz), 116.9 (dd, *J* = 19.0, 1.6 Hz), 115.9 (d, *J* = 21.4 Hz), 43.2, 38.3, 25.3; MS (*m/z*): 287 (M^+^); Anal. Calcd for C_12_H_11_F_2_NO_3_S: C, 50.17; H, 3.86; N, 4.88. Found: C, 50.02; H, 3.81; N, 4.78.

#### 3.3.13. 1-(1-(Methylsulfonyl)-1,2-dihydro-1,8-naphthyridin-3-yl)ethan-1-one (**42**)

Yield: 0.37 g (87%) as a white solid, m.p. 187–189 °C; IR: 1666, 1347, 1154 cm^−1^; ^1^H NMR (400 MHz, CDCl_3_): δ 8.31 (dd, *J* = 5.0, 1.8 Hz, 1H), 7.55 (dd, *J* = 7.5, 1.8 Hz, 1H), 7.25 (s, 1H), 7.04 (dd, *J* = 7.5, 5.0 Hz, 1H), 4.79 (d, *J* = 1.4 Hz, 2H), 3.50 (s, 3H), 2.43 (s, 3H); ^13^C NMR (101 MHz, CDCl_3_): δ 195.0, 151.4, 149.2, 137.1, 134.4, 132.7, 119.54, 119.49, 43.8, 42.2, 25.3; MS (*m/z*): 252 (M^+^); Anal. Calcd for C_11_H_12_N_2_O_3_S: C, 52.37; H, 4.79; N, 11.10. Found: C, 52.28; H, 4.77; N, 11.01.

### 3.4. General Procedure for Elimination of the Methylsulfonyl Group to Give Quinolines

To a solution of the dihydroquinoline (1 mmol) in DMF (2 mL) under N_2_, K_2_CO_3_ (1 mmol) was added, and the reaction was stirred at 80–90 °C for 2 h. At this time, TLC (20% EtOAc/hexane) indicated the complete elimination of the methane- or *p*-toluenesulfonyl group. The solution was poured into water (15 mL) and extracted with EtOAc (3 × 25 mL). The organic layer was washed with 1 M HCl (2 × 25 mL), saturated NaHCO_3_ (2 × 25 mL) and saturated NaCl (30 mL) and then dried (Na_2_SO_4_). Removal of the solvent under vacuum gave a crude product that was further purified via column chromatography to afford the quinoline product.

#### 3.4.1. Ethyl 6-Nitroquinoline-3-carboxylate (**43**)

Yield: 2.11 g (83%) as a white solid, m.p. 186–187 °C (lit. [31] m.p. 186–187 °C); IR: 1715, 1520, 1354 cm^−1^; ^1^H NMR (400 MHz, CDCl_3_): δ 9.62 (d, *J* = 2.2 Hz, 1H), 9.03 (d, *J* = 2.2 Hz, 1H), 8.92 (d, *J* = 2.5 H, 1H), 8.59 (dd, *J* = 9.2, 2.5 Hz, 1H), 8.32 (d, *J* = 9.2 Hz, 1H), 4.53 (q, *J* = 7.1 Hz, 2H), 1.49 (t, *J* = 7.1 Hz, 3H); ^13^C NMR (101 MHz, CDCl_3_): δ 164.4, 153.4, 151.6, 146.1, 140.1, 131.5, 125.9, 125.6, 125.1, 125.0, 62.1, 14.3; MS (*m/z*): 246 (M^+^). Elimination of the *p*-toluenesulfonyl group afforded this same quinoline in 81% yield.

#### 3.4.2. Ethyl 6-Cyanoquinoline-3-carboxylate (**44**)

Yield: 0.29 g (86%) as a white solid, m.p. 138–139 °C; IR: 2229, 1713 cm^−1^; ^1^H NMR (400 MHz, CDCl_3_): δ 9.58 (d, *J* = 2.1 Hz, 1H), 8.90 (d, *J* = 2.1 Hz, 1H), 8.36 (d, *J* = 1.8 Hz, 1H), 8.26 (d, *J* = 8.7 Hz, 1H), 7.97 (dd, *J* = 8.7, 1.8 Hz, 1H), 4.52 (q, *J* = 7.1 Hz, 2H), 1.49 (t, *J* = 7.1 Hz, 3H); ^13^C NMR (101 MHz, CDCl_3_): δ 164.5, 152.9, 150.5, 138.7, 135.2, 132.2, 131.2, 126.3, 124.9, 118.0, 111.4, 62.0, 14.3; MS (*m/z*): 226 (M^+^); Anal. Calcd for C_13_H_10_N_2_O_2_: C, 69.02; H, 4.46; N, 12.38. Found: C, 68.95; H, 4.41; N, 12.27.

#### 3.4.3. Ethyl 6-(Trifluoromethyl)quinoline-3-carboxylate (**45**)

Yield: 0.27 g (87%) as a white solid, m.p. 94–96 °C; IR: 1718 cm^−1^; ^1^H NMR (400 MHz, CDCl_3_): δ 9.56 (d, *J* = 2.2 Hz, 1H), 8.94 (d, *J* = 2.2 Hz, 1H), 8.28 (d, *J* = 9.0 Hz, 1H), 8.27 (s, 1H), 8.00 (dd, *J* = 9.0, 2.2 Hz, 1H), 4.51 (q, *J* = 7.1 Hz, 2H), 1.48 (t, *J* = 7.1 Hz, 3H); ^13^C NMR (101 MHz, CDCl_3_): δ 164.8, 152.2, 150.7, 139.3, 130.8, 129.4 (q, *J* = 33.0 Hz), 127.4 (q, *J* = 3.0 Hz), 127.0 (q, *J* = 4.4 Hz), 126.0, 124.5, 123.7 (q, *J* = 272.4 Hz), 61.9, 14.3; MS (*m/z*): 269 (M^+·^); Anal. Calcd for C_13_H_10_F_3_NO_2_: C, 58.00; H, 3.74; N, 5.20. Found: C, 57.94; H, 3.72; N, 5.07.

#### 3.4.4. Ethyl 6-Chloroquinoline-3-carboxylate (**46**)

Yield: 0.27 g (96%) as a white solid, m.p. 136–138 °C; IR 1717 cm^−1^; ^1^H NMR (400 MHz, CDCl_3_): δ 9.56 (d, *J* = 1.9 Hz, 1H), 8.94 (s, 1H), 8.29 (d, *J* = 9.0 Hz, 1H), 8.27 (s, 1H), 8.00 (d, *J* = 9.0 Hz, 1H), 4.51 (q, *J* = 7.1 Hz, 2H), 1.48 (t, *J* = 7.1 Hz, 3H); ^13^C NMR (101 MHz, CDCl_3_): δ 165.0, 150.3, 148.2, 137.6, 133.3, 132.6, 131.1, 127.6, 127.5, 124.1, 61.7, 14.3; HRMS (*m/z*): 235 (M^+·^); Anal. Calcd for C_12_H_10_ClNO_2_: C, 61.16; H, 4.28; N, 5.94. Found: C, 61.05; H, 4.24; N, 5.87.

#### 3.4.5. 6-Nitroquinoline-3-carbonitrile (**48**)

Yield: 0.72 g (95%) as a light yellow solid, m.p. 202–203 °C; IR: 2331, 1528, 1329 cm^−1^; ^1^H NMR (400 MHz, CDCl_3_): δ 9.22 (d, *J* = 2.0 Hz, 1H), 8.88 (d, *J* = 2.5 Hz, 1H), 8.75 (dd, *J* = 2.0, 1.0 Hz, 1H), 8.66 (ddd, *J* = 9.3, 2.5, 1.0 Hz, 1H), 8.36 (d, *J* = 9.3 Hz, 1H); ^13^C NMR (101 MHz, CDCl_3_): δ 152.8, 150.5, 146.7, 142.9, 132.0, 126.0, 125.2, 124.7, 116.0, 108.9; MS (*m/z*): 199.0382 (M^+^); Anal. Calcd for C_10_H_5_N_3_O_2_: C, 60.31; H, 2.53; N, 21.10. Found: C, 60.35; H, 2.59; N, 21.12.

### 3.5. Base Promoted Sulfone Migration in 32 to Give Ethyl 7-(Methylsulfonyl)-6-(phenethyl-amino)quinoline-3-carboxylate (**49**)

To a mixture of 0.10 g (0.11 mL, 0.85 mmol) of phenethylamine and 0.12 g (0.85 mmol) of K_2_CO_3_ in anhydrous DMF (2 mL) at 23 °C, 0.27 g (0.85 mmol) of **32** was added. The reaction was stirred for 1 h at 23 °C and for 12 h at 90 °C, after which TLC (20% EtOAc/hexane) indicated that the reactants were consumed. The reaction was cooled, added to water (20 mL) and extracted with ethyl acetate (3 × 15 mL). The combined organic extracts were washed with water (25 mL) and saturated NaCl (25 mL) and then dried (Na_2_SO_4_). Concentration under vacuum and purification via PTLC afforded 0.32 g (95%) of **49** as a yellow solid, m.p. 153–154 °C; IR: 3385, 1721, 1283, 1140 cm^−1^; ^1^H NMR (400 MHz, CDCl_3_): δ 9.22 (d, *J* = 2.1 Hz, 1H), 8.65 (s, 1H), 8.63 (dd, *J* = 2.1, 1.0 Hz, 1H), 7.37–7.23 (complex, 5H), 7.00 (s, 1H), 6.15 (br t, *J* = 5.0 Hz, 1H), 4.48 (q, *J* = 7.1 Hz, 2H), 3.55 (dt, *J* = 6.9, 5.0 Hz, 2H), 3.08 (t, *J* = 6.9 Hz, 2H), 2.90 (s, 3H), 1.46 (t, *J* = 7.1 Hz, 3H); ^13^C NMR (101 MHz, CDCl_3_): δ 165.1, 147.5, 143.4, 142.3, 138.7, 136.4, 132.9, 131.8, 130.5, 128.80, 128.76, 126.8, 125.7, 106.9, 61.7, 44.9, 42.1, 34.9, 14.3; MS (*m/z*): 398 (M^+^); Anal. Calcd for C_21_H_22_N_2_O_4_S: C, 63.30; H, 5.57; N, 7.03. Found: C, 63.26; H, 5.55; N, 6.99. A crystal (Appendix A) of **49** for X-ray structure determination was obtained from EtOH and was filed at the CCDC under deposition number 2246638.

## 4. Conclusions

A series of new MBH acetates were prepared and reacted with methanesulfonamide to produce tertiary dihydroquinoline sulfonamides in high yields. One example using *p*-toluenesulfonamide was also performed to further demonstrate the scope of the reaction. Both ester- and ketone-derived MBH acetates gave excellent yields of the cyclic sulfonamides, whereas several of the acrylonitrile-derived substrates failed to undergo the final cyclization. Further efforts to eliminate the methylsulfonyl group from these derivatives as a route to quinolines met with varying results. Although dihydroquinoline sulfonamides derived from ethyl acrylate (with one exception) and acrylonitrile (NO_2_-activated only) generally underwent elimination to give excellent yields of quinolines, those derived from 3-buten-2-one failed to undergo the elimination and decomposed. This problem presumably derives from the enolizable methyl ketone at C-3 of the dihydroquinolines. Further experiments on the acrylate-derived dihydroquinoline **32** bearing fluorines at C-6 and C-7 indicated that other reaction processes could occur in preference to the desired elimination. Treatment of this dihydroquinoline with phenethylamine gave 1,3-sulfonyl migrated amine **49** in 95% yield. This rearrangement is most often promoted by metals and acids, but it occurred under base conditions in the current work. Two mechanisms are proposed. Although the exact mechanism cannot be definitively assigned, the favored pathway involves (1) an attack at C-6 to displace the N-1 mesyl anion; (2) an attack of this species at C-7 to give the imine-stabilized anion; (3) the elimination of fluoride from C-7; (4) the base-promoted rearomatization of the system with the elimination of fluoride from C-6. The alternative mechanism involving a S_N_Ar addition–elimination through both rings is disfavored due to the weak activation of the system by the ester and the presence of an electron-donating amine ortho at the site of nucleophilic attack. Further work is in progress to elucidate the details of this interesting process.

## Data Availability

Not applicable.

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
