# Peer review of "Synthesis and Elimination Pathways of 1-Methanesulfonyl-1,2-dihydroquinoline Sulfonamides"

_molecules, 2023, doi:10.3390/molecules28073256_

Round 1

Reviewer 1 Report

Additional comments:

The mechanism of the reaction for the ring closure by TsNH2 or MsNH2 should be shown. At present, it is not clear to me how the nitrogen closes on the aromatic ring after the initial Michael addition or SN2’ substitution to the MBH acetate adduct.

Author Response

I appreciate the comments made by Reviewer 1. I have inserted Scheme 1 which outlines the reaction (with some mechanistic details) that we hoped to achieve. I have also added Scheme 2 which shows the previous copper-catalyzed coupling work by the Wang group. I attempted to clarify the substitution patterns on the various derivatives by adding number schemes on some of the structures and specifying the substitution on the substrates. The idea of adding multiple letters after each number to indicate substituted groups seemed confusing.

Though we still favor mechanism 2, we have made a statement that we cannot definitively determine which mechanism is operating in the 1,3-sulfonyl migration reaction.

Reviewer 2 Report

The manuscript reported a series of tertiary dihydroquinoline sulfonamides and further aromatization under mild basic conditions to deliver quinolines with interesting sulfonyl migration in the 6,7-di-F derivative. The experimental work has been done well, and the paper is well-written and easily understandable. Therefore, I recommend accepting it after minor revision. The detailed comments are as follows:

  1. Abstract/Tables 1-2: Reaction temperature is 23°C or room temperature?
  2. Introduction: It is better to have some background of the Morita-Baylis-Hillman (MBH) and aza-Michael reactions for the general readers.
  3. Figure 1. For a better understanding, tertiary cyclic sulfonamides moiety can be highlighted with different colors or bold in b/w for the compounds 1-6. 
  4. Authors have mentioned that: Acrylonitrile substrates 13-15, however, failed to undergo this reaction, an observation likely due to the diminished reactivity of unsaturated nitriles in the Michael-initiated SN2' reaction…What is the driving force for having compound 27 in excellent yield (96%), whereas it failed to give compounds 28-30 with the same acrylonitrile substrates? Authors should provide the experimental data of compounds 28-30 (initial addition products).
  5. The formation of 1,3-sulfonyl migrated amine 43 is quite interesting; therefore, to generalize this, it would be great to check how the reaction proceeds with other halogens and different N-R (R= different N-protecting groups).
  6. Mechanism: sulfonyl anion (mesyl group) migration has been proposed in two different pathways; what if you use the N-tosyl group with 6,7-di-F substitutes?
  7. Supporting information: The author should provide the NMR plots (NMR peak values should be identical) as mentioned in the experimental section; it needs replotting the spectra, and better to have expanded 1H peaks.

Author Response

This review had many useful suggestions and several corrections for which we are grateful.

  1. In our work, we prefer to specify the temperature (23 oC) rather than use the less specific "room temperature".
  1. By inserting Schemes 1 and 2, we have more clearly outlined our project goals and the work which has already been reported.
  1. We have highlighted the tertiary cyclic sulfonamide in blue for each compound.
  2. We have attempted to address the failure to form compounds 13-15 (which are now 35-37) in the text of the manuscript. These compounds gave TLC results that suggested that some initial Michael addition occurred, but these gave complex mixtures upon attempted isolation and decomposition on prolonged heating. The NO2 group is by far the best activating group for the SNAr reaction. CN is usually a good activator as well, but we could not push all of these to the final product. CF3 and Cl are weaker activating groups and we were not surprised that these did not provide cyclized products.
  1. Further studies of the 1,3-sulfonyl migration are currently underway. We will report these in due course.
  1. Other work has focused on migrations of N-tosyl and N-phenyl groups. It is highly likely that these will also migrate in our substrates, but we have not pursued these reactions in this project. This may be one of the first cases of a methanesulfonyl group migration, however.
  1. We have reported our NMR data as we always do. We have expanded all of the spectra for our interpretations. From spectrum to spectrum, chemical shifts are sometimes ±0.1-0.2 ppm different (sometimes more for H-bonded protons) in the 1H NMRs due to concentration effects and the polarity of the solvent.